# Electroacupuncture for acute postoperative pain during coughing after video-assisted thoracoscopic surgery: study protocol for a pilot randomized controlled trial

**Dan-Tong Zhang[1], Guang-Xia Shi[1], Jin-Ying Jia[2], Feng-Wei Tan[3], Yu Qi[2], Wei Liang[2], Zhi-Jun Zhao[2], Ping Yuan[2], Ya Gao[2], Guo-Chao Zhang[3], Xue-Hui Liu[4], Jing-Wen Yang[1], Li-Qiong Wang[1], Xiao Wang[1], Na-Na Yang[1]\*, Jian-Feng Tu🆔[1]\*, Cun-Zhi Liu[1]**

**1** International Acupuncture and Moxibustion Innovation Institute, School of Acupuncture-Moxibustion and Tuina, Beijing University of Chinese Medicine, Beijing, China, **2** The First Affiliated Hospital of Zhengzhou University, Zhengzhou, Henan, China, **3** Department of Thoracic Surgery, National Cancer Center/National Clinical Research Center for Cancer/Cancer Hospital, Chinese Academy of Medical Sciences and Peking Union Medical College, Beijing, China, **4** Medical College, Tianjin University, Tianjin, China

\* 1254614551@qq.com (NNY); tujianfeng1@126.com (JFT)

## Abstract

### Background

Despite the use of multimodal analgesia regimens following video-assisted thoracoscopic surgery (VATS), acute postoperative pain persists. Electroacupuncture (EA) may alleviate acute postoperative pain following VATS; however, current evidence remains insufficient.

### Objectives

This trial will assess the feasibility and efficacy of EA as an adjunctive treatment compared with sham electroacupuncture (SA) for managing acute postoperative pain during coughing after VATS.

### Study design and methods

In this pilot randomized controlled trial, 60 patients with non-small-cell lung cancer who will undergo VATS will be randomly assigned to either the EA or SA group in a 1:1 ratio. Patients will receive standard care combined with four sessions of EA or SA within 72h after surgery. The primary outcome will be the average postoperative pain score during coughing, measured using a numeric rating scale within 72h after surgery. Secondary outcomes will include the average daily pain scores at rest, during coughing, and while mobilizing; incidence of chronic post-surgical pain; quality of life; cough symptom score; and analgesic consumption. All adverse events will be recorded in detail.

### Discussion

This study will determine whether EA as an adjunctive treatment can reduce acute postoperative pain during coughing following VATS.

**Data availability statement:** No datasets were generated or analysed during the current study. All relevant data from this study will be made available upon study completion.

**Funding:** This study is supported by the Qihuang Scholar Project of the National Administration of Traditional Chinese Medicine. Cun-Zhi Liu is the holder of this fund. Besides, this fund has no grant numbers.

**Competing interests:** The authors have declared that no competing interests exist.

**Abbreviations:** VATS, Video-assisted thoracoscopic surgery; NSCLC, Non-small-cell lung cancer; EA, Electroacupuncture; SA Sham electroacupuncture; NRS, 11-point Numeric Rating Scale; CPSP Chronic post-surgical pain; PCIA, Patient-controlled intravenous analgesia; ITMCTR, International Traditional Medicine Clinical Trial Registry; AEs, Adverse events; EQ-5D-5L, five-level EuroQol five-dimensional questionnaire; CSS, Cough symptom score; CRF, Case report form.

## Trial registration

ClinicalTrial.gov ITMCTR2024000170 (http://itmctr.ccebtcm.org.cn/zh-CN/Home/ProjectView?pid=f1a344f7-7e06-4665-ad9d-0917d588eba6).

## Introduction

In thoracic surgery, acute postoperative pain is a common complication after video-assisted thoracoscopic surgery (VATS) which has been widely used to treat non-small-cell lung cancer (NSCLC). Doctors will encourage patients to expectorate sputum to prevent pulmonary complications after surgery. However, pain following VATS may lead to patients being unwilling to cough and expectorate sputum [1]. Therefore, reducing postoperative pain during coughing has significant implications for patients' recovery after VATS.

With the rapid development of minimally invasive technology, compared to thoracotomy, VATS is associated with less trauma, lower postoperative pain levels, better quality of life, shorter length of hospitalization, and lower complication rates [2,3]. However, acute postoperative pain during coughing remains a considerable issue after VATS. Approximately 51.7% of patients experienced moderate to severe pain when coughing within the first 24 hours after surgery [4]. On postoperative day 7, around 20% of patients still encountered moderate and severe pain while coughing [5]. Notably, inadequate management of acute postoperative pain frequently results in chronic post-surgical pain (CPSP). Therefore, it is urgent to find effective therapies to relieve acute postoperative pain and prevent it transforming into CPSP.

Due to the remarkable analgesic effect, acupuncture has been widely used to alleviate various clinical pains such as acute renal colic [6], migraine [7], and low back pain [8,9]. Recent research indicated that acupuncture may alleviate acute pain after VATS [10,11]. However, the results were limited by the poor methodological quality. It is unclear whether acupuncture can alleviate acute postoperative pain after VATS. We designed this randomized controlled trial to determine whether electroacupuncture (EA) as an adjunctive treatment compared with sham electroacupuncture (SA) can reduce acute postoperative pain during coughing following VATS.

## Methods and analysis

### Study design and setting

This pilot randomized controlled trial will include a treatment period of 3 days and a follow-up period of 3 months. The study will be conducted in the inpatient wards of two hospitals in China (The First Affiliated Hospital of Zhengzhou University and Cancer Hospital Chinese Academy of Medical Sciences). Sixty patients will be recruited and randomly assigned to the EA or SA group in a 1:1 ratio. The schedule of enrolment, intervention, and assessment is presented in Fig 1. The flow diagram of the trial is illustrated in Fig 2. The protocol follows the guidelines set by the Standard Protocol Items: Recommendations for Interventional Trials (SPIRIT) [12], and Standards for Reporting Interventions in Clinical Trials of Acupuncture (STRICTA) [13]. This study has been registered with the International Traditional Medicine Clinical Trial Registry (ITMCTR2024000170) on August 5, 2024.

### Ethical considerations

All participants will provide written informed consent before randomization. The protocol (version 2.0, January 16, 2024) has been approved by the ethics committees of the Beijing University of Chinese Medicine (No. 2024BZYLL0407). The findings of this trial will be published in peer-reviewed journals. The private information of patients will be kept strictly confidential. Participants will be allowed to withdraw from the study at any time.

| | Enrolment | Allocation | Post-allocation | | | | Closeout |
|---|---|---|---|---|---|---|---|
| **TIME POINT** | | | 0h | 24h | 48h | 72h | month 1 | month 3 |
| **Enrolment** | | | | | | | | |
| Eligibility screen | ✕ | | | | | | | |
| Informed consent | ✕ | | | | | | | |
| VATS | ✕ | | | | | | | |
| Randomisation | | ✕ | | | | | | |
| **Intervention** | | | | | | | | |
| EA | | | ✕ | ✕ | ✕ | ✕ | | |
| SA | | | ✕ | ✕ | ✕ | ✕ | | |
| **Assessment** | | | | | | | | |
| NRS pain at coughing | | | | ✕ | ✕ | ✕ | | |
| NRS pain at rest | | | | ✕ | ✕ | ✕ | | ✕ |
| NRS pain at mobilization | | | | ✕ | ✕ | ✕ | | |
| The CPSP rate | | | | | | | | ✕ |
| EQ-5D-5L | ✕ | | | | ✕ | | ✕ | ✕ |
| Cough symptom score | | | | | | | ✕ | ✕ |
| Analgesic consumption | | | ●———————————————————● | | | | | |
| First expectorated sputum | | | ●———————————● | | | | |
| Chest tube duration | | | ●———————————————————————————● | | | | |
| Length of postoperative hospital stay | | | ●———————————————————————————● | | | | |
| Pulmonary complications | | | ●———————————————————————————● | | | | |
| Blinding assessment | | | | ✕ | | | | |
| Adverse events | | | ●———————————————————————————● | | | | |

**Fig 1. Schedule of enrollment, intervention, and assessment.** Schedule of enrolment, intervention, and assessment. VATS: video-assisted thoracoscopic surgery, EA: electroacupuncture, SA: sham electroacupuncture, NRS: numerical rating scale, CPSP: chronic post-surgical pain, EQ-5D-5L: five-level EuroQol five-dimensional questionnaire.

## Participants

Patients with NSCLC scheduled to undergo VATS for pulmonary resection will be informed about the trial before surgery. All patients will receive general anesthesia during the surgery. If a patient expresses interest in participating in the trial, the clinical research coordinator (CRC) will contact them to record their basic information and assess eligibility based on the inclusion and exclusion criteria.

## Inclusion criteria

1. Men or women aged 18 to 75 years

2. Patients diagnosed with NSCLC before or during surgery

3. Undergoing VATS for pulmonary resection (including lung wedge resection, segmentectomy, and pulmonary lobectomy) for the first time

4. American Society of Anesthesiologists physical status I or II

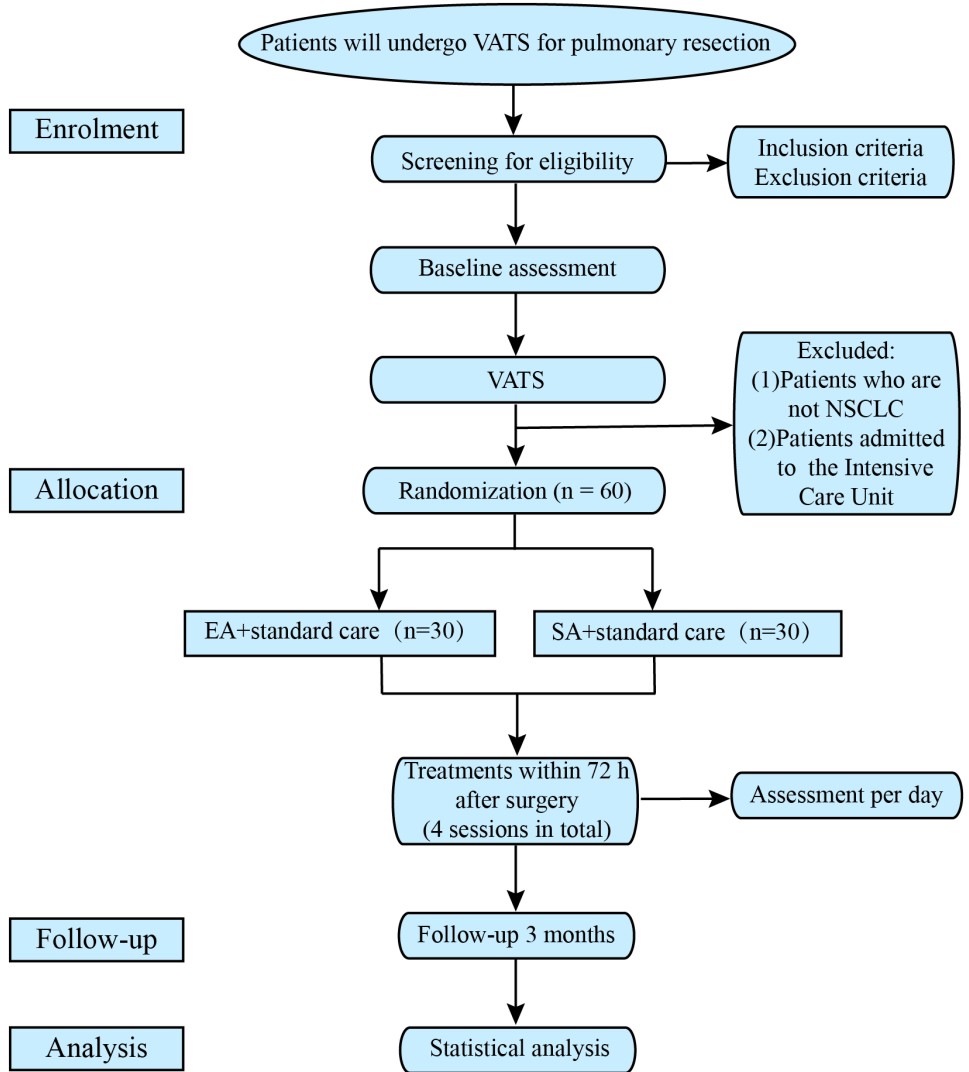

**Fig 2. Study flow diagram.** VATS: video-assisted thoracoscopic surgery, NSCLC: non-small cell lung cancer, EA: electroacupuncture, SA: sham electroacupuncture.

5. Having no communication barriers and can cooperate with 3 months telephone follow-up

6. Willing to sign informed consent

## Exclusion criteria

1. Patients with a history of previous chest surgery (e.g., mastectomy, thoracotomy, or thoracoscopic surgery) or chronic pain

2. Patients with a history of preoperative respiratory infection, chronic cough, sputum production, or postnasal drip syndrome

3. Having received chemotherapy, radiation therapy, or acupuncture treatment within 3 months prior to the trial

4. Conversion to open surgery or require postoperative admission to the intensive care unit

5. Patients who are allergic to analgesic drugs and unwilling to use patient-controlled intravenous analgesia (PCIA) after surgery

6. Planning to receive additional surgeries within 3 months postoperatively

7. Patients with a history of alcohol abuse or drug misuse

8. Having neurological illnesses, pregnancy, lactation, severe hypertension, diabetes, cardiovascular disease, serious hepatic or renal insufficiency, or infectious diseases

9. Having metal allergy, coagulation abnormalities, infections at the selected acupoint sites, or the presence of implantable medical devices such as pacemakers

10. Participation in other clinical trials

## Randomization and Blinding

Sixty eligible patients will be randomly assigned to the EA or SA group in a 1:1 ratio. Stratified block randomization will be applied with stratification according to the two research centers and variable block lengths. A statistician not involved in the implementation or statistical analysis of the trial will generate the random sequences using the STATA statistical software (version 12.0; StataCorp LP). The CRC will contact an independent administrator by phone to obtain randomization information. The acupuncturists will be informed of the group allocation after surgery. The study procedure is outlined in Fig 3. To reduce potential bias, patients, outcome assessors, and the statistician will be blinded to the group assignments.

## Procedure

The details of each surgery will be documented. All patients will receive PCIA for postoperative pain management. The analgesic solution, totalling 200 ml, will be prepared by the physician based on the patient's condition. If the numeric rating scale (NRS) pain score at rest is ≥ 4 after two consecutive PCIA [1], additional dezocine will be administered intravenously as rescue medication.

Acupuncture will be performed when the patients return to the ward after surgery. Licensed acupuncturists with a minimum of three years of acupuncture experience will conduct EA and SA treatments. Before the trial, all acupuncturists will be trained in standardized operating procedures, including acupuncture manipulation, accurate locations of acupoints and non-acupoints, and operation of an EA apparatus. Treatment offered in this

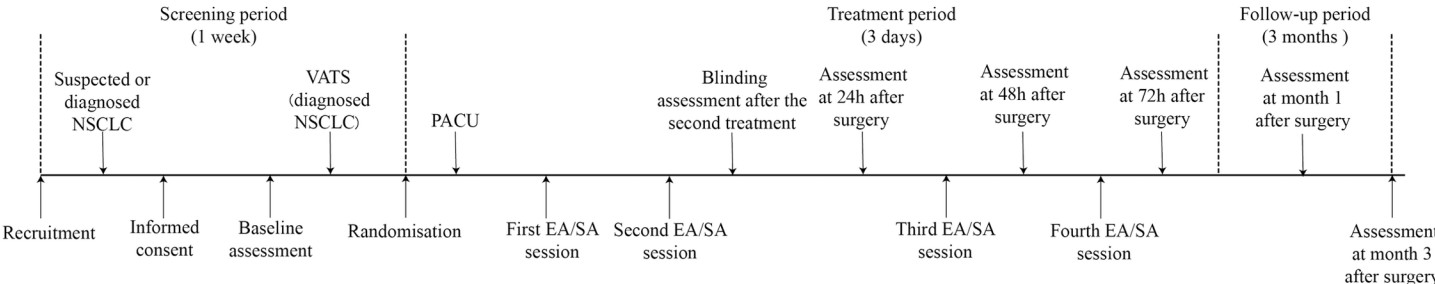

**Fig 3. Study procedure.** VATS: video-assisted thoracoscopic surgery, NSCLC: non-small cell lung cancer, PACU: post-anesthesia care unit, EA: electroacupuncture, SA: sham electroacupuncture.

trial, developed by experts, is based on previous studies [10,14]. Patients will receive four sessions of EA or SA within 72 h after surgery (treatment frequency decreasing from 2 sessions for the first 24 h to 1 session per 24 h for the remaining hours). Each treatment session will last 30 min. Acupuncturists will be required to maintain minimal communication with patients.

## Electroacupuncture

According to traditional Chinese medicine meridian theory, the bilateral *Taichong* (LR3), *Yanglingquan* (GB34), *Kongzui* (LU6), *Neiguan* (PC6), *Hegu* (LI4), and *Neimadian* (extra points) will be identified and marked by acupuncturists. The acupoint locations are listed in Table 1 and Fig 4. All the acupoints will be disinfected with alcohol. Single-use sterile acupuncture needles (Hwato disposable acupuncture needle; Suzhou, Jiangsu, China) will be used. Needles measuring 0.25 mm × 40 mm will be inserted to a depth of approximately 10–20 mm at LR3, GB34, LU6, PC6, and LI4. *Neimadian* will be inserted with needles of 0.40 mm × 50 mm to a depth of approximately 30 mm. After the needles are inserted into the skin at an angle of 90°, twirling, lifting, and thrusting manipulations will be applied to reach *de qi* sensation (a sensation combining sourness, numbness, distention, and heaviness). Electric stimulation will be performed for 30 min with an EA apparatus (Electronic Acupuncture Treatment Instrument, Suzhou Medical Co., Ltd.). The paired electrodes of the EA apparatus will be attached to the needle handles at GB34 and *Neimadian*. Continuous waves at a frequency of 2 Hz will be applied, gradually increasing the electric current to a tolerable level [14].

## Sham electroacupuncture

SA will be used as the control to verify the efficacy of EA in this trial. Six non-acupoints are selected for the SA group. The locations of the non-acupoints are shown in Fig 4 and Table 2. The needling depth will be 2–3 mm. Non-acupoint 5 will be inserted with needles of 0.40 mm × 25 mm. Needles of 0.25 mm × 25 mm will be inserted at other non-acupoints. There will be no manipulation to reach *de qi* at any non-acupoint. The appearance of the EA apparatus will be identical in both groups. However, the internal wires of the EA apparatus will be cut in the SA group. Hence, no electric current will be delivered while the apparatus will be turned on. The paired electrodes of the EA apparatus will be attached to the needle handles at non-acupoint 4 and non-acupoint 5. The other treatment steps will be the same as the EA group.

**Table 1. Location of acupoints for EA group.**

| Acupoints | Locations[a] |
|---|---|
| *Taichong* (LR3) | On the dorsum of the foot, between the first and second metatarsal bones, in the depression distal to the junction of the bases of the two bones, over the dorsalis pedis artery |
| *Yanglingquan* (GB34) | On the fibular aspect of the leg, in the depression anterior and distal to the head of the fibula |
| *Kongzui* (LU6) | On the anterolateral aspect of the forearm, on the line connecting LU5 with LU9, 7 cun superior to the palmar wrist crease |
| *Neiguan* (PC6) | On the anterior aspect of the forearm, between the tendons of the palmaris longus and the flexor carpi radialis, 2 cun proximal to the palmar wrist crease |
| *Hegu* (LI4) | On the dorsum of the hand, radial to the midpoint of the second metacarpal bone |
| *Neimadian* (Extra point) | On the medial side of the lower leg, 7 cun above the medial malleolus and about 0.5 cun from post edge of the tibia |

Abbreviations: LR: Liver, GB: Gallbladder, LU: Lung, PC: Pericardium Channel, LI: Large Intestine, EA: Electroacupuncture.

[a] 1 cun (≈20 mm) is defined as the width of the interphalangeal joint of the patient's thumb

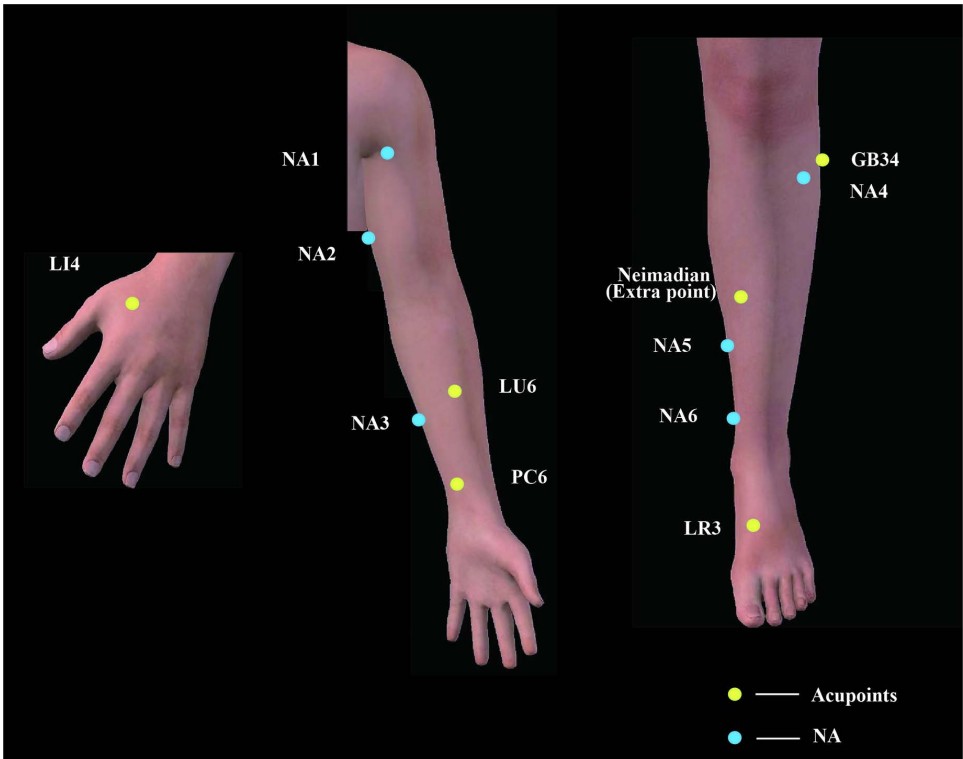

**Fig 4. Locations of acupoints and non-acupoints.** NA: non-acupoints. Reprinted from [3Dbody application] under a CC BY license, with permission from [3Dbody (Shanghai) Digital Technology Co., Ltd.], original copyright [2024].

## Outcomes

**Primary outcome.** The primary outcome is the average postoperative pain score during coughing, within 72 h after surgery. Pain scores will be measured using the NRS at 24, 48, and 72 h after surgery; the average of these scores will represent the average postoperative pain score. The higher the NRS score (range, 0-10), the more severe the pain. The validated NRS has always been used to measure various clinical pain [15].

**Table 2. Location of non-acupoints for SA group.**

| Non-Acupoints | Locations |
|---|---|
| Non-Acupoint 1 | On the front arm of deltoid muscle and biceps brachi junction |
| Non-Acupoint 2 | Half way between the tip of the elbow and the axilla |
| Non-Acupoint 3 | on the ulnar side of the arm, half way between the epicondylus medialis of the humerus and the ulnar side of the wrist |
| Non-Acupoint 4 | In the middle of *Yanglingquan* (GB34) and *Zusanli* (ST36) points (between the gallbladder and stomach meridian) |
| Non-Acupoint 5 | 2 cun above Sanyinjiao (SP6), between the liver and spleen meridian |
| Non-Acupoint 6 | 2 cun above the medial malleolus, between the liver and spleen meridian |

Abbreviations: GB: Gallbladder, ST: Stomach, EX: Extra points, UE: Upper Extremities, SP: Spleen, SA: Sham electroacupuncture.

[a]1 cun (≈20 mm) is defined as the width of the interphalangeal joint of the patient's thumb.

**Secondary outcomes. Pain severity.** Pain severity during rest, coughing, and mobilization will be measured using the NRS at 24, 48, and 72 h after VATS. Additionally, pain scores at rest will be recorded at month 3 after VATS. Besides, the responder rate will be measured at 72h; a responder is defined as a participant with an NRS score ≤ 4 when coughing.

**Chronic post-surgical pain.** The incidence of CPSP will be defined as NRS ≥ 1 at month 3 after surgery [16].

**Quality of life.** Quality of life will be assessed using a five-level EuroQol five-dimensional questionnaire (EQ-5D-5L) before surgery and at 24h, 72h, and month 3 after surgery [17]. The EQ-5D-5L assesses five dimensions (mobility, self-care, usual activities, pain/discomfort, and anxiety/depression), with each dimension divided into five levels from 1 to 5. Higher levels indicate a lower quality of life.

**Cough symptom score.** The severity of cough will be assessed using the cough symptom score (CSS) at month 1 and month 3 after surgery. The CSS includes daytime and nighttime cough symptoms, with scores ranging from 0 (no cough) to 3 (most severe cough). These scores will be added to determine the overall cough symptom score. Higher total scores indicate more severe symptoms [18].

**Analgesic consumption.** The dosage of the PCIA will be recorded throughout the trial. The use of rescue medications will also be documented in the case report form (CRF).

Additionally, postoperative pulmonary complications, first expectorated sputum, chest tube duration, and length of postoperative hospital stay will be recorded.

**Blinding assessment.** After the second EA/SA treatment, all patients will be asked to answer the question, 'Which type of acupuncture do you think you received, EA or Micro-electroacupuncture or Unclear?'. The James' blinding index ranges from 0 to 1, whereby 0 corresponds to no blinding, 0.5 corresponds to completely random blinding, and 1 corresponds to complete blinding [19]. The Bang's blinding index ranges from -1 to 1, with 1 indicating complete lack of blinding, 0 indicating perfect blinding, and -1 indicating opposite guessing [20].

**Adverse events.** During the study period, all adverse events (AEs) will be recorded daily in the CRF. The AEs associated with acupuncture include infection, hematoma, fainting, broken needles, and continuous post-needling pain. The drug-associated AEs include nausea, vomiting, and allergic reactions. Serious adverse events are defined as requiring hospitalization, causing persistent or severe disability, threatening life, or resulting in death. Serious adverse events will be reported to the principal investigator within 24 h.

## Data management

Data will be recorded in the CRF and entered into the database twice to ensure accuracy. After completion of the data audit, the database will be locked by the data management team, and data can no longer be modified. Patient information, including name, ID card number, medical history, and laboratory results, will be confidential. Both paper and electronic documents will be retained for at least 5 years after publication.

## Quality control

Experts in acupuncture, thoracic surgery, methodology, and statistics have reviewed the trial protocols. Central randomization will be used to control for bias. Before the start of the trial, all researchers will receive unified training that includes participant recruitment, random allocation, acupuncture manipulation, and communication with patients. A unified CRF will be used for recording, and the researcher will monthly monitor the trial.

## Sample size calculation

Based on previous studies [21], the average postoperative pain score during coughing for the EA and SA groups is estimated to be 3.0 ± 1.7 and 4.4 ± 1.7. To achieve a statistical power of 80% with a two-sided significance level of 5%, 24 patients per group are required. Sixty patients (30 per group) are required, accounting for a dropout rate of 20%. This pilot trial is intended to establish initial data for the primary outcome measure, and the results will be used to calculate the sample size for the next larger randomized controlled trial.

## Statistical analysis

All randomly assigned patients will be included in an intention-to-treat analysis. Multiple imputations will be used for missing data. Normally distributed continuous variables will be described as mean ± standard deviations (M ± SD) and analyzed using independent t-test. Skewed continuous variables will be represented as medians combined with inter-quartile ranges and analyzed using the Wilcoxon rank-sum test. Categorical data will be analyzed using the chi-squared test and presented as frequencies, percentages, or constituent ratios.

Sensitivity analyses will be conducted for the primary outcome in the per-protocol analysis, which includes participants completing 80% of the treatment without major protocol deviations or omissions. Besides, to examine the robustness of the result, the primary outcome will be performed using a linear regression model as sensitivity analysis adjusted for sex, age, smoking history at baseline, and analgesic consumption, chest tube duration, the number of ports. We will perform a predefined subgroup analysis of primary outcome according to chest tube duration ($< 3/\geq 3$ days), and the number of ports (1/2/3).

The James' blinding index and the Bang's blinding index will be used to assess the adequacy of blinding. Statistical analyses will be conducted by a statistician using the SPSS Statistics 27. $P < 0.05$ will be considered statistically significant.

# Discussion

Acute postoperative pain often occurs after various surgical procedures. Presently, there are limitations to the postoperative pain management of VATS. This study aims to assess whether EA may provide an adjunctive treatment for acute postoperative pain management after VATS. Additionally, we will observe whether EA can be integrated relatively easily into the existing postoperative pain management of VATS.

In clinical practice, despite multimodal analgesia (e.g., PCIA, paravertebral block, and oral medication), pain during coughing remains a frequent complaint in the early postoperative period after VATS. The mechanism of acupuncture for pain has been partially revealed [22,23]. EA can relieve pain, reduces the use of analgesic drugs, and has fewer AEs in patients undergoing thoracotomy [14]. In this trial, EA will be added to the existing analgesic methods to enhance the analgesic effect in patients undergoing VATS. The design of this study is based on clinical practice, which can reflect the analgesic effect of EA in complex clinical situations and identify situations where EA may be more suitable.

LR3, GB34, LU6, PC6, LI4, and Neimadian (extra points) have been used to reduce postoperative pain in patients undergoing thoracic surgery [10,14,24,25]. According to the traditional Chinese medicine meridian theory and clinical experience, we finally determined the treatment acupoints for this study. Allocation concealment and blinding that meet the methodological demands will be used to control bias and ensure the accuracy of the results. All researchers will have a division of labor and will not cross each other. This pilot study will provide the necessary information for a full-scale randomized controlled trial. This study

recruitment period is expected between September 2024 and June 2025. The trial is planned to be completed in December 2025.

This proposed trial has some limitations. First, although the sample size is calculated, it is relatively small and susceptible to influence by other factors, resulting in bias. Second, the acupuncturists can not be masked because of the characteristics of acupuncture. However, the other investigators in our study will be blinded.

## Supporting information

**S1 File. SPIRIT checklist.**
(PDF)

**S2 File. Original protocol approved by the ethics committee.**
(PDF)

**S3 File. Original protocol approved by the ethics committee (Chinese).**
(PDF)

## Acknowledgments

We appreciate the contribution of all the researchers and patients involved in this study.

## Author contributions

**Conceptualization:** Dan-Tong Zhang, Jian-Feng Tu, Cun-Zhi Liu.

**Formal analysis:** Jian-Feng Tu.

**Investigation:** Dan-Tong Zhang, Jin-Ying Jia, Feng-Wei Tan, Jian-Feng Tu.

**Supervision:** Jin-Ying Jia, Feng-Wei Tan, Zhi-Jun Zhao, Jian-Feng Tu, Cun-Zhi Liu.

**Writing – original draft:** Dan-Tong Zhang.

**Writing – review & editing:** Dan-Tong Zhang, Guang-Xia Shi, Jin-Ying Jia, Feng-Wei Tan, Yu Qi, Wei Liang, Zhi-Jun Zhao, Ping Yuan, Ya Gao, Guo-Chao Zhang, Xue-Hui Liu, Jing-Wen Yang, Li-Qiong Wang, Xiao Wang, Na-Na Yang, Jian-Feng Tu, Cun-Zhi Liu.

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
