## [Decision Letter · Decision Letter 0]

9 Oct 2024

PONE-D-24-42025Electroacupuncture for acute postoperative pain after video-assisted thoracoscopic surgery: study protocol for a pilot randomized controlled trialPLOS ONE

Dear Dr. Tu,

Thank you for submitting your manuscript to PLOS ONE. After careful consideration, we feel that it has merit but does not fully meet PLOS ONE’s publication criteria as it currently stands. Therefore, we invite you to submit a revised version of the manuscript that addresses the points raised during the review process.

We look forward to receiving your revised manuscript.

Kind regards,

Hantong Hu

Academic Editor

PLOS ONE

Journal Requirements:

4. We note that Figure 4 in your submission contain copyrighted images. All PLOS content is published under the Creative Commons Attribution License (CC BY 4.0), which means that the manuscript, images, and Supporting Information files will be freely available online, and any third party is permitted to access, download, copy, distribute, and use these materials in any way, even commercially, with proper attribution. For more information, see our copyright guidelines: http://journals.plos.org/plosone/s/licenses-and-copyright.

a. You may seek permission from the original copyright holder of Figure 4  to publish the content specifically under the CC BY 4.0 license.

Reviewers' comments:

Reviewer's Responses to Questions

**Comments to the Author**

1. Does the manuscript provide a valid rationale for the proposed study, with clearly identified and justified research questions?

Reviewer #1: Yes

Reviewer #2: Yes

2. Is the protocol technically sound and planned in a manner that will lead to a meaningful outcome and allow testing the stated hypotheses?

Reviewer #1: Partly

Reviewer #2: Yes

3. Is the methodology feasible and described in sufficient detail to allow the work to be replicable?

Reviewer #1: No

Reviewer #2: Yes

4. Have the authors described where all data underlying the findings will be made available when the study is complete?

Reviewer #1: No

Reviewer #2: Yes

5. Is the manuscript presented in an intelligible fashion and written in standard English?

Reviewer #1: No

Reviewer #2: Yes

6. Review Comments to the Author

You may also provide optional suggestions and comments to authors that they might find helpful in planning their study.

Reviewer #1: This manuscript reports on the protocol of a randomised sham-controlled trial of electroacupuncture for acute postoperative pain after video-assisted thoracoscopic surgery. While this manuscript provides details of study plans, there seem to be several points to be clarified or modified.

1)As the authors calculated a sample size and defined a primary outcome to conduct a formal statistical analysis to evaluate effect size of the intervention, it is not a pilot study. It’s not a pilot study simply because the sample size is small.

2)Since it is a sham-controlled trial protocol, I would like the authors to report on detailed wording given to participants regarding sham. Were the participants aware of the existence of sham electroacupuncture and that they may be randomly allocated to either real or fake intervention? This information about whether the term ‘fake’, ‘placebo’, ‘dummy’ or ‘sham’ was used in the informed consent process is important in the aspects of participant blinding and the direction of trial outcomes.

3)As per author guidelines of this journal, authors of protocols without data are strongly encouraged to state how they plan to share their data when it is completed or published. Stating ‘No datasets were generated or analysed during the current study. All relevant data from this study will be made available upon study completion’ does not give any information how data will be shared. Please state your plan how to share your data. Are your going to use publicly accessable link, or provide supplementary files of raw data/analytic codes?

4)I would like to reconsider “average” postoperative pain as a primary outcome. Usually pain intensity data measured with NRS or VAS are not normally distributed. It means that it is common that you get (almost) complete pain relief or (almost) no help at all. It is highly likely that you belong to either end of the NRS, good or bad. So if you use the “mean” data, you are using nobody’s result. So in acute pain trials, it is appropriate to use responder/non-responder analysis and I recommend you reconsider adopting this approach and recalcuating the sample size accordingly.

5)Please report datails about how you will assess participant blinding. Why are you going to evaluate blinding after the ‘2nd’ treatment, not after 1st or at the end of treatment? What is your rationale for selecting ‘after the 2nd treatment’? How many options are your going to give? Real or sham? Or Real, sham, or don’t know? Are you going to calculate blinding index and what statistical method are you going to use? All these information will be useful and you’ve got a plenty of space for such information as this is a protocol paper.

6)Page 5, lines 79-81: "It is unclear that the efficacy of electroacupuncture in alleviating acute postoperative pain after VATS." It's grammatically incorrect.

Reviewer #2: It is interesting that authors designed a clinical trial to evaluate the efficacy of electroacupuncture in acute postoperative pain during coughing after video-assisted thoracoscopic surgery, I think several concerns listed below will need to be addressed to improve paper quality.

1. The primary outcome is the average postoperative pain score during coughing. Please revise the title to match the primary outcome.

2. Please provide the rationale for the treatment acupoints selection in the discussion section.

3. The primary outcome is the average postoperative pain score during coughing, measured using a numeric rating scroe in the Abstract section(line 36-38, page 2). However, it is measured using the PCIA in the Outcomes section (line 242- 245). Which is correct?

4. When is the evaluation time point for the primary endpoint? (at 24h or 48h or 72h after surgery) Please clarify it.

5. Is the primary outcome the changes in NRS at each timepoint? Is the secondary outcome the changes in secondary outcome at each timepoint? Please clarify it.

7. PLOS authors have the option to publish the peer review history of their article (what does this mean? ). If published, this will include your full peer review and any attached files.

**Do you want your identity to be public for this peer review?** For information about this choice, including consent withdrawal, please see our Privacy Policy .

Reviewer #1: **Yes: ** Hyangsook Lee

Reviewer #2: No

---

## [Author Response · Author response to Decision Letter 0]

18 Nov 2024

Reviewer: 1

As the authors calculated a sample size and defined a primary outcome to conduct a formal statistical analysis to evaluate effect size of the intervention, it is not a pilot study. It’s not a pilot study simply because the sample size is small.

Response: Thank you for your valuable comments. Your perspective aligns with many scholars. However, it is indeed a topic of debate whether to calculate the sample size in the pilot study. Several scholars think the justification of the sample size for a pilot study is necessary and the primary outcome should be defined in the pilot study [1-3]. The sample size has been calculated in some published pilot studies [4,5]. Therefore, we cursorily calculated the sample size for this study. We are able to obtain relatively stable data from the pilot study, which is crucial for preventing under powering in future full-scale trial.

References

1. Kunselman AR. A brief overview of pilot studies and their sample size justification. Fertility and sterility. 2024;121(6):899-901. https://doi.org/10.1016/j.fertnstert.2024.01.040 PMID: 38331310

2. Fayers PM, Machin D. Sample size: how many patients are necessary? British journal of cancer. 1995;72(1):1-9. https://doi.org/10.1038/bjc.1995.268 PMID: 7599035

3. Wang Y, Wang LQ, Yang JW, Shi GX, Tu JF, Yan SY, et al. Application and reflection on pilot study in acupuncture clinical research. Zhongguo Zhen Jiu. 2021;41(3):325-9. https://doi.org/10.13703/j.0255-2930.20201024-0002 PMID: 33798319.

4. Lin LL, Tu JF, Wang LQ, Yang JW, Shi GX, Li JL, et al. Acupuncture of different treatment frequencies in knee osteoarthritis: a pilot randomised controlled trial. Pain. 2020;161(11):2532-8. https://doi.org/10.1097/j.pain.0000000000001940 PMID: 32483056

5. Swartz HA, Rollman BL, Mohr DC, Sadow S, Frank E. A randomized pilot study of Rhythms And You (RAY): An internet-based program for bipolar disorder administered with and without clinical helper support in primary care. Journal of affective disorders. 2021;295:183-91. https://doi.org/10.1016/j.jad.2021.08.025

-Since it is a sham-controlled trial protocol, I would like the authors to report on detailed wording given to participants regarding sham. Were the participants aware of the existence of sham electroacupuncture and that they may be randomly allocated to either real or fake intervention? This information about whether the term ‘fake’, ‘placebo’, ‘dummy’ or ‘sham’ was used in the informed consent process is important in the aspects of participant blinding and the direction of trial outcomes.

Response: Thank you for your suggestions. After reviewing the checklist, we discovered an issue with the version of the uploaded informed consent. We apologize for our mistake and have resubmitted the informed consent. The above words are not mentioned in the informed consent but are replaced by micro-electroacupuncture to maximize the effectiveness of blinding. The partial disclosure is often used in the study of acupuncture [1]. Sufficient blinding will increase the credibility of the measured outcomes [2]. We will report the blind index to assess the adequacy of blinding.

References

1. Foster NE, Thomas E, Barlas P, Hill JC, Young J, Mason E, et al. Acupuncture as an adjunct to exercise based physiotherapy for osteoarthritis of the knee: randomised controlled trial. BMJ (Clinical research ed). 2007;335(7617):436. https://doi.org/10.1136/bmj.39280.509803.BE PMID: 17699546

2. Lee YS, Kim SY, Lee H, Chae Y, Lee MS. ACURATE: A guide for reporting sham controls in trials using acupuncture. Journal of evidence-based medicine. 2023;16(1):82-90. https://doi.org/10.1111/jebm.12524 PMID: 36959765

-As per author guidelines of this journal, authors of protocols without data are strongly encouraged to state how they plan to share their data when it is completed or published. Stating ‘No datasets were generated or analysed during the current study. All relevant data from this study will be made available upon study completion’ does not give any information how data will be shared. Please state your plan how to share your data. Are your going to use publicly accessable link, or provide supplementary files of raw data/analytic codes?

Response: Thank you for your valuable suggestions. We apologize for our negligence. We have stated the plan for sharing data at Lines 414-415 in the manuscript as “It is permitted for reviewers or readers to access the original data by contacting the corresponding author (Jian-Feng Tu)”.

-I would like to reconsider “average” postoperative pain as a primary outcome. Usually pain intensity data measured with NRS or VAS are not normally distributed. It means that it is common that you get (almost) complete pain relief or (almost) no help at all. It is highly likely that you belong to either end of the NRS, good or bad. So if you use the “mean” data, you are using nobody’s result. So in acute pain trials, it is appropriate to use responder/non-responder analysis and I recommend you reconsider adopting this approach and recalculating the sample size accordingly.

Response: Thank you for your suggestions. The primary outcome in this study is the global pain state during 72h, which is calculated by the average of the NRS scores at 24h, 48h, and 72h. This similar primary outcome is also used in other acute pain studies [1-3]. According to your opinion, we have added the response as the secondary outcome as follows “Besides, the responder rate will be measured at 72h; a responder is defined as a participant with an NRS score ≤ 4 when coughing” (at Lines 264-266). The NRS score of 4 is the threshold for tolerable postoperative pain [4].

References

1. Ilfeld BM, Plunkett A, Vijjeswarapu AM, Hackworth R, Dhanjal S, Turan A, et al. Percutaneous Peripheral Nerve Stimulation (Neuromodulation) for Postoperative Pain: A Randomized, Sham-controlled Pilot Study. Anesthesiology. 2021;135(1):95-110. https://doi.org/10.1097/ALN.0000000000003776 PMID: 33856424

2. Jiwnani S, Ranganathan P, Patil V, Agarwal V, Karimundackal G, Pramesh CS. Pain after posterolateral versus nerve-sparing thoracotomy: A randomized trial. J Thorac Cardiovasc Surg. 2019; 157:380-6. https://doi.org/10.1016/j.jtcvs.2018.07.033 PMID: 30195601

3. Machado LA, Maher CG, Herbert RD, Clare H, McAuley JH. The effectiveness of the McKenzie method in addition to first-line care for acute low back pain: a randomized controlled trial. BMC medicine. 2010;8:10. https://doi.org/10.1186/1741-7015-8-10 PMID: 20102596

4. Gerbershagen HJ, Rothaug J, Kalkman CJ, Meissner W. Determination of moderate-to-severe postoperative pain on the numeric rating scale: a cut-off point analysis applying four different methods. British journal of anaesthesia. 2011;107(4):619-26. https://doi.org/10.1093/bja/aer195 PMID: 21724620.

-Please report datails about how you will assess participant blinding. Why are you going to evaluate blinding after the ‘2nd’ treatment, not after 1st or at the end of treatment? What is your rationale for selecting ‘after the 2nd treatment’? How many options are your going to give? Real or sham? Or Real, sham, or don’t know? Are you going to calculate blinding index and what statistical method are you going to use? All these information will be useful and you’ve got a plenty of space for such information as this is a protocol paper.

Response: Thank you for your kind advices. Evaluation of blindness after the first treatment will not predict the integrity of blinding in subsequent phase of the trial [1]. If blinding is evaluated at the end of treatment, participants may judge the grouping according to the clinical effectiveness [2]. Therefore, we evaluate blinding after the ‘2nd’ treatment. We have added the sentence at Lines 353-354 in the manuscript as “The James' blinding indices will be used to assess the adequacy of blinding.” We will give participants three options. A sentence has been added to the manuscript as “After the second EA/SA treatment, all patients will be asked to answer the question, ‘Which type of acupuncture do you think you received, EA or Micro-electroacupuncture or Unclear?’”. (Lines 297-300)

References

1. Hopton AK, Macpherson H. Assessing blinding in randomised controlled trials of acupuncture: challenges and recommendations. Chinese journal of integrative medicine. 2011;17(3):173-6. https://doi.org/10.1007/s11655-011-0663-9 PMID: 21359917

2. Sackett DL. Turning a blind eye: why we don't test for blindness at the end of our trials. BMJ (Clinical research ed). 2004;328(7448):1136. https://doi.org/10.1136/bmj.328.7448.1136-a PMID: 15130997

-Page 5, lines 79-81: "It is unclear that the efficacy of electroacupuncture in alleviating acute postoperative pain after VATS." It's grammatically incorrect.

Response: Thank you for your valuable suggestions. We have modified the sentence at Lines 81-82 as “It is unclear whether EA can alleviate acute postoperative pain after VATS.”

Reviewer: 2

-The primary outcome is the average postoperative pain score during coughing. Please revise the title to match the primary outcome.

Response: Thank you for your professional suggestions. We have revised the title to match the primary outcome as “Electroacupuncture for acute postoperative pain during coughing after video-assisted thoracoscopic surgery: study protocol for a pilot randomized controlled trial”.

-Please provide the rationale for the treatment acupoints selection in the discussion section.

Response: Thank you for your comments. We have added the rationale for the treatment acupoints selection as “LR3, GB34, LU6, PC6, LI4, and Neimadian (extra points) have been used to reduce postoperative pain in patients undergoing thoracic surgery [14, 15, 23, 24]. According to the traditional Chinese medicine meridian theory and clinical experience, we finally determined the treatment acupoints for this study” at Line 377-381.

-The primary outcome is the average postoperative pain score during coughing, measured using a numeric rating scroe in the Abstract section(line 36-38, page 2). However, it is measured using the PCIA in the Outcomes section (line 242- 245). Which is correct?

Response: Thank you for your valuable suggestions. We apologize for our mistake. We have modified the sentence at Line 256 as “Pain scores will be measured using the NRS at 24, 48, and 72 h after surgery”.

-When is the evaluation time point for the primary endpoint? (at 24h or 48h or 72h after surgery) Please clarify it.

-Is the primary outcome the changes in NRS at each timepoint? Is the secondary outcome the changes in secondary outcome at each timepoint? Please clarify it.

Response: Thank you for your suggestions. The evaluation time point for the NRS is at 24h, 48h, and 72h after surgery. The primary outcome is the average of pain scores evaluated at 24h, 48h, and 72h after surgery. Pain score during coughing of NRS at each timepoint is a secondary outcome.

---

## [Editor Report · Decision Letter 1]

29 Nov 2024

PONE-D-24-42025R1Electroacupuncture for acute postoperative pain during coughing after video-assisted thoracoscopic surgery : study protocol for a  pilot randomized controlled trialPLOS ONE

Dear Dr. Tu,

Thank you for submitting your manuscript to PLOS ONE. After careful consideration, we feel that it has merit but does not fully meet PLOS ONE’s publication criteria as it currently stands. Therefore, we invite you to submit a revised version of the manuscript that addresses the points raised during the review process.

We look forward to receiving your revised manuscript.

Kind regards,

Hantong Hu

Academic Editor

PLOS ONE

Journal Requirements:

Additional Editor Comments:

The comments of two reviewers have been addressed, and the quality of the manuscript has been significantly improved, but there are still two issues that need further elaboration.

1. The article only briefly mentions the assessment of whether blinding was successful for the patients: "The James' blinding indices will be used to assess the adequacy of blinding." Please provide more specific relevant details and references of the James' blinding indices. Notably, the Bang's blinding index is also a commonly used method in the field of sham acupuncture to assess whether patient blinding was successful. Please clarify the differences between the two methods and explain why you ultimately chose the James' blinding indices over the Bang's blinding index.

2. In the revised manuscript, "a responder is defined as a participant with an NRS score ≤ 4 when coughing" (at Lines 264-266). However, the baseline NRS score of patients will also impact the post-treatment score. Why not define a responder based on a certain threshold of change in NRS score before and after treatment?

---

## [Author Response · Author response to Decision Letter 1]

17 Dec 2024

Editor Comments:

-The article only briefly mentions the assessment of whether blinding was successful for the patients: "The James' blinding indices will be used to assess the adequacy of blinding." Please provide more specific relevant details and references of the James' blinding indices. Notably, the Bang's blinding index is also a commonly used method in the field of sham acupuncture to assess whether patient blinding was successful. Please clarify the differences between the two methods and explain why you ultimately chose the James' blinding indices over the Bang's blinding index.

Response: Thank you for your valuable comments. James' blinding index is different from Bang's blinding index. James' blinding index combines blinding data from all arms, but Bang's blinding index assesses blinding separately for different arms [1]. Therefore, we will use James' and Bang's blinding index to assess the adequacy of blinding (at Lines 356-357). We have added the sentence at Lines 298-303 in the manuscript: “The James' blinding index ranges from 0 to 1, whereby 0 corresponds to no blinding, 0.5 corresponds to completely random blinding, and 1 corresponds to complete blinding [19]. The Bang's blinding index ranges from -1 to 1, with 1 indicating complete lack of blinding, 0 indicating perfect blinding, and -1 indicating opposite guessing [20].”

References

1. Bang H, Ni L, Davis CE. Assessment of blinding in clinical trials. Controlled clinical trials. 2004;25(2):143-56. doi: 10.1016/j.cct.2003.10.016. PubMed PMID: 15020033.

-In the revised manuscript, "a responder is defined as a participant with an NRS score ≤ 4 when coughing" (at Lines 264-266). However, the baseline NRS score of patients will also impact the post-treatment score. Why not define a responder based on a certain threshold of change in NRS score before and after treatment?

Response: Thank you for your suggestions. The NRS was not evaluated before treatment in this study. Due to the varying efficacy of anesthetics on patients, the time for patients to be fully awake after surgery is different. If patients are not fully awake, we think that the NRS scores they give are inaccurate. Hence, we don’t evaluate the NRS scores before treatment, which is similar to other studies [1-3].

References

1. Ibrahim M, Menna C, Andreetti C, Puyo C, Maurizi G, D'Andrilli A, et al. Does a Multimodal No-Compression Suture Technique of the Intercostal Space Reduce Chronic Postthoracotomy Pain? A Prospective Randomized Study. Journal of thoracic oncology. 2016;11(9):1460-8. doi: 10.1016/j.jtho.2016.05.023. PubMed PMID: 27282308.

2. Ding W, Chen Y, Li D, Wang L, Liu H, Wang H, et al. Investigation of single-dose thoracic paravertebral analgesia for postoperative pain control after thoracoscopic lobectomy - A randomized controlled trial. International journal of surgery. 2018;57:8-14. doi: 10.1016/j.ijsu.2018.07.006. PubMed PMID: 30056127.

3. Jiwnani S, Ranganathan P, Patil V, Agarwal V, Karimundackal G, Pramesh CS. Pain after posterolateral versus nerve-sparing thoracotomy: A randomized trial. The Journal of thoracic and cardiovascular surgery. 2019;157(1):380-6. doi: 10.1016/j.jtcvs.2018.07.033. PubMed PMID: 30195601.

---

## [Decision Letter · Decision Letter 2]

30 Dec 2024

PONE-D-24-42025R2Electroacupuncture for acute postoperative pain during coughing after video-assisted thoracoscopic surgery : study protocol for a  pilot randomized controlled trialPLOS ONE

Dear Dr. Tu,

Thank you for submitting your manuscript to PLOS ONE. After careful consideration, we feel that it has merit but does not fully meet PLOS ONE’s publication criteria as it currently stands. Therefore, we invite you to submit a revised version of the manuscript that addresses the points raised during the review process.

We look forward to receiving your revised manuscript.

Kind regards,

Hantong Hu

Academic Editor

PLOS ONE

Journal Requirements:

Additional Editor Comments:

As requested by the Editorial Office of Plos One, this paper must include a review by a statistical expert. Please make minor revisions to the manuscript based on this statistical expert’s comments.

Reviewers' comments:

Reviewer's Responses to Questions

**Comments to the Author**

1. Does the manuscript provide a valid rationale for the proposed study, with clearly identified and justified research questions?

Reviewer #3: Yes

2. Is the protocol technically sound and planned in a manner that will lead to a meaningful outcome and allow testing the stated hypotheses?

Reviewer #3: Partly

3. Is the methodology feasible and described in sufficient detail to allow the work to be replicable?

Reviewer #3: Yes

4. Have the authors described where all data underlying the findings will be made available when the study is complete?

Reviewer #3: Yes

5. Is the manuscript presented in an intelligible fashion and written in standard English?

Reviewer #3: Yes

6. Review Comments to the Author

You may also provide optional suggestions and comments to authors that they might find helpful in planning their study.

Reviewer #3: The document is well written. There are a few data issues that should be clarified. For example, the quality control section appears to be directed to the researchers. The investigators state that the researcher will regularly monitor the trial. What exactly is ,'regularly'? Also in the pain assessment, the patient will rate their pain which can be somewhat subjective. How will this assessment be validated? This should at least be mentioned in the protocol.

Concerning the sample size and analysis, the sample size appears adequate for a pilot effort. For comparative univariate statistical analysis, the t-test and Wilcoxon appear to suffice as needed. Are there any thoughts of any adjustment (multivariate perhaps) to the analysis based on relevant patient characteristics or clinical variables?

Also , presumably statements such as ,” We will perform a predefined subgroup analysis of primary endpoints according to chest tube duration and the number of incisions.”, will be detailed in the protocol.

7. PLOS authors have the option to publish the peer review history of their article (what does this mean? ). If published, this will include your full peer review and any attached files.

**Do you want your identity to be public for this peer review?** For information about this choice, including consent withdrawal, please see our Privacy Policy .

Reviewer #3: No

---

## [Author Response · Author response to Decision Letter 2]

20 Jan 2025

Reviewer: 3

- The document is well written. There are a few data issues that should be clarified. For example, the quality control section appears to be directed to the researchers. The investigators state that the researcher will regularly monitor the trial. What exactly is, 'regularly'?

Response: Thank you for your suggestions. 'Regularly' means that we will conduct monthly monitoring of the trial. [1]. We have revised the sentence at Line 329 in the manuscript as “A unified CRF will be used for recording, and the researcher will monthly monitor the trial.”

References

1. Wu XK, Gao JS, Ma HL, Wang Y, Zhang B, Liu ZL, et al. Acupuncture and Doxylamine-Pyridoxine for Nausea and Vomiting in Pregnancy : A Randomized, Controlled, 2 × 2 Factorial Trial. Annals of internal medicine. 2023;176(7):922-33. doi: 10.7326/m22-2974. PubMed PMID: 37335994.

-Also in the pain assessment, the patient will rate their pain which can be somewhat subjective. How will this assessment be validated? This should at least be mentioned in the protocol.

Response: Thank you for your valuable comments. In our study, postoperative pain will be evaluated by the validated NRS [1]. Besides, the NRS has always been used in many high-quality clinical studies [2-4]. We have added this sentence to the manuscript (at Lines 257-258).

References

1. Ferreira-Valente MA, Pais-Ribeiro JL, Jensen MP. Validity of four pain intensity rating scales. Pain. 2011;152(10):2399-404. doi: 10.1016/j.pain.2011.07.005. PubMed PMID: 21856077.

2. Malfliet A, Kregel J, Coppieters I, De Pauw R, Meeus M, Roussel N, et al. Effect of Pain Neuroscience Education Combined With Cognition-Targeted Motor Control Training on Chronic Spinal Pain: A Randomized Clinical Trial. JAMA neurology. 2018;75(7):808-17. doi: 10.1001/jamaneurol.2018.0492. PubMed PMID: 29710099.

3. Tankha H, Gaskins D, Shallcross A, Rothberg M, Hu B, Guo N, et al. Effectiveness of Virtual Yoga for Chronic Low Back Pain: A Randomized Clinical Trial. JAMA network open. 2024;7(11):e2442339. doi: 10.1001/jamanetworkopen.2024.42339. PubMed PMID: 39485352.

4. Bendixen M, Jørgensen OD, Kronborg C, Andersen C, Licht PB. Postoperative pain and quality of life after lobectomy via video-assisted thoracoscopic surgery or anterolateral thoracotomy for early stage lung cancer: a randomised controlled trial. The Lancet Oncology. 2016;17(6):836-44. doi: 10.1016/s1470-2045(16)00173-x. PubMed PMID: 27160473.

- Concerning the sample size and analysis, the sample size appears adequate for a pilot effort. For comparative univariate statistical analysis, the t-test and Wilcoxon appear to suffice as needed. Are there any thoughts of any adjustment (multivariate perhaps) to the analysis based on relevant patient characteristics or clinical variables?

Response: Thank you for your comments. We will use a linear regression model as the sensitivity analysis to examine the robustness of the result. Based on the previous research, we found that some factors impact the primary outcome such as sex, age, smoking history, analgesic consumption, chest tube duration, and the number of ports [1-3]. We have added the sentence at Lines 354-358 in the manuscript as “Besides, to examine the robustness of the result, the primary outcome will be performed using a linear regression model as sensitivity analysis adjusted for sex, age, smoking history at baseline, and analgesic consumption, chest tube duration, the number of ports.”

References

1. Sun K, Liu D, Chen J, Yu S, Bai Y, Chen C, et al. Moderate-severe postoperative pain in patients undergoing video-assisted thoracoscopic surgery: A retrospective study. Scientific reports. 2020;10(1):795. doi: 10.1038/s41598-020-57620-8. PubMed PMID: 31964955.

2. Yun T, Zhang Y, Liu A, Qin Y, Sun X, Wu Z, et al. Randomized Trial of Modified Chest Tube Placement vs Routine Placement After Lung Resection. The Annals of thoracic surgery. 2023;116(5):1013-9. doi: 10.1016/j.athoracsur.2023.04.031. PubMed PMID: 37146783.

3. Tamura M, Shimizu Y, Hashizume Y. Pain following thoracoscopic surgery: retrospective analysis between single-incision and three-port video-assisted thoracoscopic surgery. Journal of cardiothoracic surgery. 2013;8:153. doi: 10.1186/1749-8090-8-153. PubMed PMID: 23759173.

-Also, presumably statements such as,” We will perform a predefined subgroup analysis of primary endpoints according to chest tube duration and the number of incisions.”, will be detailed in the protocol.

Response: Thank you for your valuable suggestions. Previous studies found that the average chest tube duration was about 3 days and the number of ports affected postoperative pain [1-2]. We have revised the sentence at Lines 358-360 in the manuscript as “We will perform a predefined subgroup analysis of primary outcome according to chest tube duration (< 3/≥3 days), and the number of ports (1/2/3).

References

1. Yun T, Zhang Y, Liu A, Qin Y, Sun X, Wu Z, et al. Randomized Trial of Modified Chest Tube Placement vs Routine Placement After Lung Resection. The Annals of thoracic surgery. 2023;116(5):1013-9. doi: 10.1016/j.athoracsur.2023.04.031. PubMed PMID: 37146783.

2. Tamura M, Shimizu Y, Hashizume Y. Pain following thoracoscopic surgery: retrospective analysis between single-incision and three-port video-assisted thoracoscopic surgery. Journal of cardiothoracic surgery. 2013;8:153. doi: 10.1186/1749-8090-8-153. PubMed PMID: 23759173.

---

## [Decision Letter · Decision Letter 3]

31 Jan 2025

Electroacupuncture for acute postoperative pain during coughing after video-assisted thoracoscopic surgery : study protocol for a  pilot randomized controlled trial

PONE-D-24-42025R3

Dear Dr. Tu,

We’re pleased to inform you that your manuscript has been judged scientifically suitable for publication and will be formally accepted for publication once it meets all outstanding technical requirements.

Kind regards,

Hantong Hu

Academic Editor

PLOS ONE

Additional Editor Comments (optional):

All previous comments of 3 reviewers have been properly addressed by the authors and I have also participated in the peer review of this paper. I think this paper has reached the level for publication in the current form.

Reviewers' comments:

Reviewer's Responses to Questions

**Comments to the Author**

1. Does the manuscript provide a valid rationale for the proposed study, with clearly identified and justified research questions?

Reviewer #3: Yes

2. Is the protocol technically sound and planned in a manner that will lead to a meaningful outcome and allow testing the stated hypotheses?

Reviewer #3: Yes

3. Is the methodology feasible and described in sufficient detail to allow the work to be replicable?

Reviewer #3: Yes

4. Have the authors described where all data underlying the findings will be made available when the study is complete?

Reviewer #3: Yes

5. Is the manuscript presented in an intelligible fashion and written in standard English?

Reviewer #3: Yes

6. Review Comments to the Author

You may also provide optional suggestions and comments to authors that they might find helpful in planning their study.

Reviewer #3: It appears that all comments have been addressed and the appropriate additions to the paper have been included.

I thank the investigators for the added clarification.

7. PLOS authors have the option to publish the peer review history of their article (what does this mean? ). If published, this will include your full peer review and any attached files.

**Do you want your identity to be public for this peer review?** For information about this choice, including consent withdrawal, please see our Privacy Policy .

Reviewer #3: No

---

## [Editor Report · Acceptance letter]

PONE-D-24-42025R3

PLOS ONE

Dear Dr. Tu,

I'm pleased to inform you that your manuscript has been deemed suitable for publication in PLOS ONE. Congratulations! Your manuscript is now being handed over to our production team.

Kind regards,

on behalf of

Dr. Hantong Hu

Academic Editor

PLOS ONE